# The Mechanistic Roles of Sirtuins in Breast and Prostate Cancer

**DOI:** 10.3390/cancers14205118

**Published:** 2022-10-19

**Authors:** Cosmos Ifeanyi Onyiba, Christopher J. Scarlett, Judith Weidenhofer

**Affiliations:** 1School of Biomedical Sciences and Pharmacy, College of Health, Medicine and Wellbeing, University of Newcastle, Ourimbah, NSW 2258, Australia; 2School of Environmental and Life Sciences, College of Engineering, Science and Environment, University of Newcastle, Ourimbah, NSW 2258, Australia; 3Hunter Medical Research Institute, New Lambton Heights, NSW 2305, Australia

**Keywords:** sirtuins, breast, prostate, cancer, tumor, miRNA, suppression, promotion, progression, metastasis

## Abstract

**Simple Summary:**

There are diverse reports of the dual role of sirtuin genes and proteins in breast and prostate cancers. This review discusses the current information on the tumor promotion or suppression roles of SIRT1–7 in breast and prostate cancers. Precisely, we highlight that sirtuins regulate various proteins implicated in proliferation, apoptosis, autophagy, chemoresistance, invasion, migration, and metastasis of both breast and prostate cancer. We also provide evidence of the direct regulation of sirtuins by miRNAs, highlighting the consequences of this regulation in breast and prostate cancer. Overall, this review reveals the potential value of sirtuins as biomarkers and/or targets for improved treatment of breast and prostate cancers.

**Abstract:**

Mammalian sirtuins (SIRT1–7) are involved in a myriad of cellular processes, including apoptosis, proliferation, differentiation, epithelial-mesenchymal transition, aging, DNA repair, senescence, viability, survival, and stress response. In this review, we discuss the current information on the mechanistic roles of SIRT1–7 and their downstream effects (tumor promotion or suppression) in cancers of the breast and prostate. Specifically, we highlight the involvement of sirtuins in the regulation of various proteins implicated in proliferation, apoptosis, autophagy, chemoresistance, invasion, migration, and metastasis of breast and prostate cancer. Additionally, we highlight the available information regarding SIRT1–7 regulation by miRNAs, laying much emphasis on the consequences in the progression of breast and prostate cancer.

## 1. Introduction

### Overview of Sirtuins

The silent information regulator 2 (sir2) family of proteins, simply known as sirtuins (sir-two-ins), was first discovered in yeast, where its upregulation increased lifespan [1]. Consistent with this finding, subsequent studies revealed that sirtuins promote longevity in *Caenorhabditis elegans* [2] and *Drosophila melanogaster* [3]. As members of the highly conserved class III histone deacetylase, sirtuins, in addition to their initial description as mono-adenosine diphosphate (ADP) transferases, were later found to be a nicotinamide (NAM) adenine dinucleotide (NAD)-dependent group of enzymes [4,5]. By transferring an acetyl group to the ribose moiety of ADP, sirtuins couple the deacetylation of lysine in their substrate protein with the hydrolysis of NAD+, forming 2′-O-acetyl-ADP-ribose (o-AADPR) and releasing NAM, a feedback inhibitor of sirtuins [6,7] (Figure 1).

In mammals, seven sirtuins (SIRT1–7) have been identified, which primarily function as NAD-dependent deacetylases (SIRT1–3 and SIRT5–7) [8] and ADP-ribosyl transferases (SIRT4 and 6) [9,10]. Additionally, sirtuins have been reported to function as demyristoylases (SIRT1–3 and 6) [11,12], lipoamidases (SIRT4) [13], and desuccinylases/demalonylases/deglutarylases (SIRT5) [14,15]. However, despite these functional differences, sirtuins are classified and distinguished by their subcellular location (Figure 2). SIRT1, 6, and 7 are predominantly located in the nucleus [16,17], whereas SIRT3, 4, and 5 are predominantly located in the mitochondrion [16]. Additionally, SIRT5 has been found expressed in the cytoplasm [18], while SIRT3 has been found to modulate some stress-related and nuclear-encoded mitochondrial gene expression in the nucleus [19] and can translocate from the nucleus to the mitochondrion upon cellular stress [20]. Recently, a study revealed that SIRT4 is also expressed in the cytoplasm and nucleus [21]. SIRT2 is predominantly located in the cytoplasm but can translocate to the nucleus to interact with its nuclear targets [22], especially during mitotic cell division [23]. Additionally, SIRT1 has been found to shuttle between the nucleus and cytoplasm [24,25]. According to the phylogenetic analysis of their core domains, sirtuins are placed into four classes: I (SIRT1, 2, and 3), II (SIRT4), III (SIRT5), and IV (SIRT6 and 7) [26,27].

By virtue of the above enzymatic functions, mammalian sirtuins are involved in a myriad of biological processes, including apoptosis, proliferation, differentiation, epithelial-mesenchymal transition (EMT), aging, DNA repair, senescence, viability, survival, and stress response, among other biological functions [27,28,29,30,31,32,33]. Thus, deregulation of sirtuins can be involved in perturbations found in metabolic disorders and cancer. Recently, the role of all mammalian sirtuins in metabolic disorders affecting six human tissues was studied [34]. Furthermore, sirtuins were found to play a dual role in carcinogenesis via tumor promotion and suppression [35].

Given that cancer is an increasing cause of death worldwide, numerous research efforts have been channeled toward understanding the molecular basis of the disease to develop effective treatments. In this regard, several studies have explored the role of sirtuins in the progression, migration, and metastasis of various types of cancer [28,36,37,38], and several anticancer agents that target sirtuins have been developed [39,40]. Additionally, several studies have shown the prognostic potential of sirtuins in various types of cancer [36,41,42]. Globally, cancers of the breast and prostate are among the most diagnosed types of cancer in females and males, respectively [27]. For instance, in Australia, 1 in 7 women and 1 in 6 men will be diagnosed with breast and prostate cancer, respectively, in their lifetime [28]. In the United States, cancers of the breast (31%) and prostate (27%) account for the largest proportion of newly estimated cancer cases in females and males, respectively [43]. Additionally, while the breast and prostate have unique functions, cancers occurring in these tissues share a similarity in both being hormonally driven [44] and typically having high proportions of indolent disease [45]. Thus, investigation of both prostate and breast cancer together can assist in identifying the relevance of specific gene expression differences toward cancer progression in general rather than nuanced differences between cancers arising in different tissues. Thus, in this review, we summarize and discuss the current information on the mechanistic roles of sirtuins (SIRT1–7) in breast and prostate cancer.

## 2. Mechanistic Roles of Sirtuins in Breast and Prostate Cancer

Sirtuins are reported to play a dual role in the pathogenesis of breast and prostate cancer via the targeting of different proteins, including transcription factors, signaling molecules, and catalytic proteins. Consequently, sirtuins have exhibited both tumor-suppressing and tumor-promoting effects in breast and prostate cancer cells (Figure 3). In this section, we discuss the mechanistic roles of nuclear (SIRT1, 6, and 7), cytoplasmic (SIRT2), and mitochondrial sirtuins (SIRT3, 4, and 5) in breast and prostate cancer (Table 1 and Table 2).

### 2.1. Nuclear Sirtuins in Breast Cancer

#### 2.1.1. SIRT1

Recent studies have revealed that SIRT1 is constitutively upregulated in breast cancer cells [49,50,58]. Additionally, SIRT1 has been shown to modulate different targets in breast cancer, including transcription factors, signaling molecules, and certain enzymes, showing opposing effects by its exhibition of either tumor promotion or tumor suppression. The most prominent transcription factors targeted by SIRT1 in breast cancer include forkhead box O3 (FOXO3), paired related homeobox 1 (PRRX1), and tumor suppressor proteins (p53 and p21). For instance, Mahmud et al. demonstrated that the downstream anti-proliferative activity of acetylated FOXO3, a tumor suppressor whose activity is vulnerable to posttranslational modifications, is activated upon *SIRT1* gene silencing in BT474 breast cancer cells, suggesting that SIRT1 deacetylates and inhibits FOXO3 activity to promote breast cancer [48]. They also demonstrated that *SIRT1* gene silencing increases BT474 cells’ sensitivity to lapatinib treatment, further suggesting that SIRT1 promotes BT474 cells’ resistance to lapatinib treatment. The protein expression of PRRX1, a transcription factor that induces epithelial-to-mesenchymal transition (EMT) and inhibits cancer stemness, was shown to be downregulated by *SIRT1* gene silencing in mouse 4T1 breast cancer cells [46]. Interestingly, *SIRT1* gene silencing upregulated the expression of the EMT-related protein E-cadherin and cancer stemness markers kruppel-like factor 4 (KLF4) and aldehyde dehydrogenase 1 family member A1 (ALDH1A1) [46]. Based on their findings, the authors concluded that SIRT1 deacetylates and stabilizes PRRX1 to inhibit breast cancer stemness and metastasis. Importantly, inhibition of SIRT1 by sirtinol, a classic inhibitor of sirtuins, resulted in the activation of p53 and consequently increased p21 protein levels in SK-BR-3 breast cancer cells, thus suggesting a tumor-promoting role of SIRT1 in breast cancer [53]. Moreover, SIRT1 expression has been previously shown to downregulate tumor suppressor genes, such as *cyclin G2* and *p53*, in estrogen-dependent breast cancer cells [59].

SIRT1 has been demonstrated to modulate several intracellular signaling protein molecules, including protein kinase B (PKB), B-cell lymphoma 2 (Bcl-2), metadherin (MTDH), Frizzled 7, EMT-related proteins, and the cluster of differentiation 36 (CD36), in breast cancer cells. In a study by Jin et al., SIRT1 was shown to directly interact with and phosphorylate PKB to promote subsequent downstream proliferative effects in MDA-MB-231 and BT-549 breast cancer cells [50]. To substantiate this finding, the authors further investigated Akt activities in SIRT1-knockdown breast cancer cells and found that proliferation was inhibited both in vitro and in vivo. Conversely, a previous study reported that SIRT1 represses estrogen receptor activation by deactivating Akt activity to suppress the growth of MCF-7 breast cancer cells [55]. These contradictory results may be due to genetic differences between the breast cancer cells used in the studies. However, further investigations are required to illuminate the possible reasons for these contradictions. The expression of Bcl-2, an anti-apoptotic or pro-survival protein, was markedly decreased in MCF-7 and MDA-MB-231 breast cancer cells following the inhibition of SIRT1 with sirtinol [51], which indicates that SIRT1 promotes the growth of breast cancer through the upregulation of the Bcl-2 protein.

Inhibition of SIRT1 with either EX-527 or SIRT1 siRNA resulted in the upregulation of c-Myc (a pro-oncogenic protein) and MTDH (an oncogenic protein) in MDA-MB-231 and BT-549 breast cancer cells. Further, in the presence of c-Myc siRNA, EX-527 could not upregulate MTDH in breast cancer cells [57]. As a mechanism, the authors suggested that the SIRT1-mediated inhibition of the proliferation of the breast cancer cells was due to the downregulation of c-Myc, which resulted in the downregulation of MTDH. Simmons et al. demonstrated that the inhibition of SIRT1 by both inhibitor VII and SIRT1 siRNA markedly downregulated the expression of Frizzled 7, a chronically activated oncogenic receptor in cancers [54]. This finding supports the tumor-promoting role of SIRT1 in breast cancer. Furthermore, another research group investigated the effect of SIRT1 inhibition by SIRT1 siRNA on EMT-related proteins (vimentin, snail-1, and E-cadherin) in MDA-MB-231, MDA-MB-436, and MDA-MB-468 breast cancer cell lines and found that the siRNA inhibition of SIRT1 downregulated the expression of vimentin (a protein required for cell migration, motility, and adhesion) and snail-1 (a repressor of E-cadherin) but upregulated the expression of E-cadherin (a cell-cell adhesion molecule), indicating that SIRT1 induces the invasion of breast cancer cells by modulating the EMT pathway [56]. On the contrary, SIRT1 overexpression by resveratrol (a classical activator of sirtuins) was reported to increase the expression of CD36 (a transmembrane receptor that regulates apoptosis and angiogenesis) to cytotoxic levels, thus inhibiting the proliferation of MCF-7 breast cancer cells [58]. Moreover, there is evidence that resveratrol induces toxicity in breast cancer cells through a SIRT1-dependent mechanism [108,109].

Interestingly, SIRT1 has also been demonstrated to regulate the expression of certain catalytic proteins in breast cancer. For instance, the transcription of *CYP19A1* (a gene encoding the aromatase enzyme) was downregulated in MDA-MB-231 breast cancer cells expressing aromatase following the inhibition of SIRT1 by treatment with both cambinol and inhibitor VII [52]. This finding indicates that SIRT1 promotes the growth of estrogen-dependent breast cancer through the upregulation of *CYP19A1* expression. Additionally, another study demonstrated that the knock-out of SIRT1 by shRNA reduced the proliferation, migration, and invasion of MCF-7 breast cancer cells [49]. The shRNA-mediated silencing of SIRT1 downregulated the expression of human DNA polymerase delta 1 (POLD1), a gene that encodes p125 (the catalytic domain of DNA polymerase δ) and regulates cell proliferation and the cell cycle. SIRT1 upregulated the activities of POLD1/p125 to aid the proliferation, migration, and invasion of MCF-7 breast cancer cells. Recently, a study demonstrated that CRISPR-Cas9-mediated deletion of SIRT1 promotes the release of exosomes (containing unique cargoes and hydrolases that degrade the extracellular matrix), impairs lysosomal function, and inhibits lysosomal acidification in MDA-MB-231 breast cancer cells [47]. The authors showed that the deletion of SIRT1 reduces the expression of the vacuolar-type H+ ATPase (V-ATPase) subunit responsible for proper lysosomal acidification and protein degradation. This indicates that SIRT1 upregulated V-ATPase activities to increase lysosomal function, which halted the secretion of exosomes containing hydrolases that degrade the extracellular matrix, thereby ultimately inhibiting the aggressiveness of MDA-MB-231 breast cancer cells.

#### 2.1.2. SIRT6

Although previous studies have demonstrated that SIRT6 expression is downregulated in several human cancers [110], including invasive breast tumors [111], and that SIRT6 expression positively correlates with the survival of breast cancer patients [112], there are few studies demonstrating the mechanistic effects of SIRT6 in breast cancer. Ioris et al. demonstrated that the overexpression of SIRT6 downregulates the transcription of PI3K-controlled genes, independent of its deacetylase activity, to suppress the progression and stem cell-like capacity of PyMT-induced breast cancer in mice, thus indicating a tumor suppressor role of SIRT6 in breast cancer [79]. Conversely, a study recently demonstrated that SIRT6 acts as a tumor promoter, given that its expression promoted MCF-7 cell growth as well as resistance to oxidative stress [78]. These effects were driven by SIRT6 deacetylation and activation of nicotinamide phosphoribosyl transferase and glucose-6-phosphate dehydrogenase, which increase NADH and NADPH levels, respectively, to protect against oxidative stress. Corroborating the tumor-promoting activities of SIRT6, a recent study demonstrated that SIRT6 deletion downregulated the expression and activity of pyruvate dehydrogenase and oxidative phosphorylation-related genes, suppressed the activity of respiratory complexes, and reduced the ATP/AMP ratio in MDA-MB-231 and MCF7 breast cancer cells [80]. The above findings reveal a vital role of SIRT6 in remodeling the metabolic profile to consistently provide energy and nutrients to drive the growth of breast cancer. Moreover, low SIRT6 expression was recently found to be associated with better overall survival of patients with breast cancer [113]. Notwithstanding, more studies are required to investigate the expression level of SIRT6 and its effect on the metabolic profile of breast cancer at various stages of carcinogenesis.

#### 2.1.3. SIRT7

The mRNA expression level of SIRT7 was found to be higher in patients with metastatic breast cancer [114]. More recently, high expression of SIRT7 was associated with poor prognosis of breast cancer [81,115] and positively correlated with increased M1 macrophage infiltration and exhaustion of T-cells in luminal breast cancer [115]. There are conflicting reports on the function of SIRT7 in breast cancer. In BT-549 and MDA-MB-231 breast cancer cells, SIRT7 deacetylated and destabilized SMAD4, resulting in reduced formation of the SMAD2/SMAD3-SMAD4 complex. On translocation to the nucleus, this complex activates EMT-related transcription factors (such as Snail1/2, Twist1/2, and Zeb1/2), thus facilitating the migration and invasion of cancer cells [116]. The authors also demonstrated that the deacetylation and destabilization of SMAD4 promote its degradation by β-TrCP1, a substrate recognition subunit of E3 ubiquitin ligase. Additionally, the authors corroborated their findings with knockout experiments, in which SIRT7 knockout with either shSIRT7 or siSIRT7 activated TGF-β signaling to promote EMT-related gene transcription and lung metastasis of breast cancer. Taken together, the authors demonstrated a tumor-suppressing role of SIRT7 in breast cancer. Furthermore, a recent study demonstrated that SIRT7 antagonizes TGF-β signaling and inhibits metastasis of breast cancer [81]. On the contrary, a more recent study revealed that SIRT7 promotes Adriamycin-induced metastasis in breast cancer by interacting with TIE2 [82], a tyrosine kinase receptor that induced the dormancy of breast cancer cells, resulting in increased resistance to chemotherapy [117]. Given the limited studies conducted to date and contradictory findings, more studies are required to provide insight into the modulation of other signaling proteins (including transcription factors) by SIRT7 causing either tumor-suppressing or tumor-promoting effects in breast cancer.

### 2.2. Nuclear Sirtuins in Prostate Cancer

#### 2.2.1. SIRT1

Several studies have shown that SIRT1 is highly expressed in prostate cancer [86,87,118,119], suggesting an important role for SIRT1 in its progression. Regardless of its predominant expression in prostate cancer cells, SIRT1 has been demonstrated to either promote or suppress prostate cancer via different mechanisms. Although the mechanisms were not fully elucidated, studies reported that SIRT1 promotes cell growth and chemoresistance in PC3 and DU145 prostate cancer cells [119] as well as the migration and invasion of DU145 prostate cancer cells [120]. A mechanistic study, using PC3 prostate cancer cells with active p53, or PC3 prostate cancer cells regardless of p53 status, demonstrated that SIRT1 inhibition by SIRT1 shRNA or sirtinol downregulates the deacetylation of p53 and FOXO1, thereby activating antiproliferative responses, such as apoptosis and senescence [86]. This suggests that SIRT1 expression may promote the development of prostate cancer by deacetylating and deactivating p53 and FOXO1. Moreover, using DU145 prostate cancer cells, it was previously demonstrated that SIRT1 deacetylates and deactivates FOXO1 transcription activities [121]. To further corroborate these findings, later studies demonstrated that SIRT1 deacetylates FOXO3, a tumor suppressor transcription factor that has been shown to inhibit cell proliferation and promote apoptosis [122], to promote its poly-ubiquitination and subsequent proteasomal degradation [123]. These findings suggest that SIRT1 may mediate its tumor-promoting activities in prostate cancer by preventing the tumor-suppressing function of FOXOs. Nonetheless, animal studies are required to investigate how the tumor microenvironment of prostate cancer influences the outcome of SIRT1 activities in this regard.

The tumor-promoting effects of SIRT1 in prostate cancer have been demonstrated through the regulation of EMT-related protein expression. A 2012 study revealed that SIRT1 promotes the migration and metastasis of DU145 and PC3 prostate cancer cells by downregulating the expression of the epithelial marker E-cadherin and upregulating the expression of mesenchymal markers (N-cadherin and fibronectin) and the EMT-inducing transcription factor ZEB1 [85]. Similarly, Cui et al. [84] showed that siRNA-mediated silencing of SIRT1 expression suppressed the migration and invasion of PC3 prostate cancer cells by upregulating E-cadherin and downregulating N-cadherin and vimentin expression, thus suggesting that SIRT1 activates EMT to promote the migration and metastasis of prostate cancer. Additionally, SIRT1 can promote the migration and invasion of PC3 prostate cancer cells by antagonizing p300/CBP-associated protein (PCAF)-catalyzed MPP8-K439 acetylation, thereby preventing ubiquitin-proteasome-mediated proteolysis of M-phase phosphoprotein 8 [88], a methyl-H3K9 binding protein that promotes *E-cadherin* gene silencing and EMT of tumor cells.

It is well-established that androgen receptors (AR) play a pivotal role in prostate carcinogenesis [124]. In this regard, studies have investigated the effect of SIRT1 expression on AR, as a mechanism of SIRT1 in the development and progression of prostate cancer. For instance, SIRT1 was demonstrated to deacetylate AR and suppress coactivator-induced interactions between AR amino and carboxyl termini to inhibit ligand-induced (dihydrotestosterone) AR transcription activity, thereby inhibiting the growth of AR-expressing LNCaP (lymph node carcinoma of the prostate) cells [90]. Similarly, it was also demonstrated that SIRT1 deacetylates AR and histone H3 at the promoter region of the prostate-specific antigen gene to suppress AR-mediated gene transcription, thereby inhibiting the proliferation of androgen-responsive LNCaP prostate cancer cells [89]. These findings suggest that low expression of SIRT1 is particularly beneficial for the growth of androgen-responsive prostate cancer. Conversely, androgen-independent prostate cancer cells, such as PC3 and DU145, have been shown to express higher levels of SIRT1 than androgen-dependent LNCaP prostate cancer cells [87]. This may suggest a differential role of SIRT1 in various sub-types of prostate cancer. In contrast with the above-mentioned studies, a more recent study revealed that SIRT1 promotes the progression of LNCaP prostate cancer cells by upregulating AR signaling [91]. Although this contradiction may be influenced by the cancer stage or context [125], it provides opportunities for future studies to ascertain the molecules or intermediates in AR signaling that are modulated by SIRT1 in favor of the progression of prostate cancer.

SIRT1 has been shown to modulate catalytic protein activity as a part of its mechanisms of action in prostate cancer. Notably, it has been revealed that SIRT1 deacetylates and upregulates the expression of matrix metalloproteinase-2 (MMP2) [83], a zinc-dependent endopeptidase that degrades the extracellular matrix to promote the invasion of cancer cells [126]. To corroborate their finding, the authors further demonstrated that siRNA-mediated knockdown of SIRT1 reduced MMP2 protein stability and zymographic activity and consequently inhibited the invasion of PC3 prostate cancer cells. This suggests that SIRT1 upregulates MMP2 activity to promote the progression of prostate cancer. Additionally, shRNA-mediated inhibition of SIRT1 expression induces proliferation and inhibits autophagy of DU145 and PC3 prostate cancer cells by repressing the phosphorylation of p70 ribosomal S6 kinases (S6K) and 4E-binding protein 1 [87], which are downstream effectors of the mammalian target of rapamycin 1 (mTORC1)-mediated protein synthesis regulation [127]. The authors further suggested that SIRT1 promotes the tumorigenesis of prostate cancer via SIRT1/S6K-mediated inhibition of autophagy. In other words, they speculated that the deacetylation of S6K by SIRT1 may trigger its dephosphorylation, leading to a halt in the synthesis of apoptotic or autophagic proteins. To elucidate this mechanism, further studies are required to clarify/verify their claims and ascertain the specific apoptotic or autophagic proteins regulated through this axis.

#### 2.2.2. SIRT6

Unlike in other types of cancer, including breast cancer, SIRT6 is overexpressed in prostate tumor tissues and cells (PC3, DU145, 22RV1, and LNCaP), with high SIRT6 expression correlating with poor overall survival of prostate cancer patients [96,102,103,104]. This indicates an important role of SIRT6 in prostate cancer, which needs to be fully explored. Liu et al. demonstrated that siRNA-mediated inhibition of SIRT6 induces cell cycle arrest at the sub-G1 phase, decreases Bcl-2 expression, induces apoptosis, increases DNA damage, and enhances chemotherapeutic sensitivity of PC3 and DU145 prostate cancer cells [102]. This indicates that SIRT6 promotes the survival and proliferation of prostate cancer by increasing Bcl-2 expression and preventing cell cycle arrest. To corroborate these findings, Xie et al. demonstrated that siRNA-mediated inhibition of SIRT6 reduced the EMT-protein N-cadherin and anti-EMT protein E-cadherin to promote the migration and invasion of PC3 and DU145 prostate cancer cells. Furthermore, a recent study reported that SIRT6 promotes the progression of prostate cancer by abrogating necroptosis-facilitated innate immunity [96]. Another recent study also demonstrated that SIRT6 promotes the progression of prostate cancer by negatively regulating the Wnt/β-Catenin signaling pathway, as shown by a decrease in the expression levels of c-Myc, cyclin D1, and β-Catenin [104]. To date, all studies unequivocally portray SIRT6 as a promoter of prostate cancer.

#### 2.2.3. SIRT7

SIRT7 is overexpressed in various human cancers [128], including prostate cancer [105,106], suggesting a pivotal role of SIRT7 in the progression of prostate cancer. Malik et al. observed an inverse relationship between SIRT7 and the expression of E-cadherin in highly aggressive PC3 prostate cancer cells; that is, there was a low expression of E-cadherin and a high expression of SIRT7 [107]. To verify their findings, using SIRT7-deficient PC3 prostate cancer cells, they demonstrated that the E-cadherin protein level was upregulated, while the mesenchymal marker vimentin and EMT-inducing transcription factor slug were both downregulated [107]. Taken together, these findings indicate that SIRT7 promotes the migration and metastasis of PC3 prostate cancer cells by upregulating EMT. Similarly, corroborating the above-mentioned findings, Haider et al. demonstrated that SIRT7-knockdown reduced the migration of DU145 prostate cancer cells and that SIRT7 overexpression promoted the aggressiveness and docetaxel resistance of LNCaP prostate cancer cells by triggering EMT as demonstrated by an increase in fibronectin [105]. Overall, these findings indicate that SIRT7 overexpression induces EMT to promote the aggressiveness of both androgen-dependent (LNCaP) and androgen-independent (DU145) prostate cancer cells.

More recently, Ding et al. demonstrated that SIRT7 depletion in LNCaP and 22RV1 prostate cancer cells significantly reduces the conversion of LC3B-I to LC3B-II, a vital process that results in autophagy, and that SIRT7 knockdown downregulates vimentin, slug, MMP2, and MMP9 in 22Rv1 prostate cancer cells [106]. They further demonstrated that SIRT7 expression is positively correlated with AR signaling in both prostate cancer tissues and cells (LNCaP and 22Rv1) and that SIRT7 knockdown upregulates SMAD4, a tumor suppressor protein [129], which interacts with AR to control its signaling in prostate cancer [130]. Taken together, these findings indicate that SIRT7 expression promotes the proliferation, metastasis, and androgen-induced autophagy of prostate cancer via induction of AR signaling. Moreover, it has been previously demonstrated that androgens promote the growth of prostate cancer cells via the induction of autophagy [131], thus indicating a favorable effect of autophagy in androgen-responsive prostate cancer cells.

### 2.3. Cytoplasmic Sirtuins in Breast Cancer

#### SIRT2

Studies have shown that SIRT2 expression is downregulated in breast cancer tissues compared to normal breast tissues [62,132,133]. Furthermore, SIRT2 was shown to be predominantly involved in the tumorigenesis and prognosis of breast cancer and that, depending on the tumor grade, SIRT2 acts as either a tumor suppressor or tumor promoter in breast tumors [132]. In this regard, studies have investigated the tumor-suppressing or tumor-promoting mechanisms of SIRT2 in breast cancer. Given that SIRT2 translocates to the nucleus during mitotic cell division, Kim et al. demonstrated that SIRT2 deacetylates CDH1 and CDC20 to positively regulate the activity of the anaphase-promoting complex/cyclosome (APC/C) [62], a multi-subunit protein that mediates ubiquitination of distinctly functional substrates, including Aurora-A and -B, survivin Plk1, Nek2A, securin, and cyclins-A and -B [134]. However, some of these APC/C substrates, such as Aurora A, cyclins, and Plk1, are overexpressed in human cancers and promote tumorigenesis [135,136,137]. Hypothetically, the overexpression of the APC/C substrates in human cancers may be due to the downregulated expression of SIRT2, which may result in aberrant mitotic cell division. To verify this hypothesis, Kim et al. demonstrated that SIRT2 deficiency impairs the activity of APC/C, leading to the accumulation of mitotic regulatory proteins (including Aurora A and Plk1), mitotic catastrophe, genetic instability, and, consequently, tumorigenesis [62]. Overall, their findings indicate a tumor-suppressing role of SIRT2 in breast cancer through the promotion of normal mitosis. To further support the tumor-suppressing role of SIRT2 in breast cancer, Fiskus et al. demonstrated that SIRT2 deacetylates and inhibits the peroxidase activity of peroxiredoxin-1 (an antioxidant), thereby sensitizing MCF-7 and MDA-MB-231 breast cancer cells to the cytotoxic effects of increased reactive oxygen species and DNA damage [60].

Conversely, Park et al. demonstrated that SIRT2 deacetylates the lysine 305 (K305) residue of the pyruvate kinase M2 (PKM2) isoform to direct glycolysis and promote tumor growth [63]. To corroborate their findings, they demonstrated that SIRT2 knockout by shSIRT2 RNA altered the PKM2 activity and glycolytic metabolism, thus indicating that SIRT2 ensures the growth and survival of breast cancer through the regulation of glycolysis and PKM2 activity. Additionally, using MCF10A breast cancer cells, Zhou et al. demonstrated that SIRT2 promotes the growth and aggressiveness of basal-like breast cancer by deacetylating the K116 domain of slug [61], a tumor promoter protein that, upon stabilization, facilitates tumor progression and metastasis through EMT, leading to loss of cell adhesion, as well as the enhancement of migratory and invasive properties of tumors [138]. As a mechanism, the authors demonstrated that the protein slug is stabilized by the deacetylation of its K116 domain. Moreover, the stabilization of slug has been shown to prevent its proteasomal degradation, facilitate its transcriptional repression of E-cadherin, and potentiate its anti-apoptotic and cell-invasive functions [139,140]. To corroborate their findings, Zhou et al. further demonstrated that silencing of SIRT2 with shSIRT2 causes the loss of aggressive basal-like breast cancer features in SUM149 and SUM1315 cancer cells [61]. Furthermore, a recent study demonstrated that SIRT2 deacetylase activity promotes the heterodimerization, nuclear retention, and stability of the breast cancer type I susceptibility protein (BRCA1) and BRCA1-associated RING domain protein I [64], a tumor suppressor protein involved in DNA repair and genome stability [141]. Another recent study revealed that SIRT2 positively regulates T cell differentiation, specifically the CD8+ T cells, to facilitate tumor immune response during breast cancer progression [65]. In summary, SIRT2 is highly involved in the initiation and progression of breast cancer, playing either a tumor-promoting or tumor-suppressing role. This opens further opportunities to explore the distinct roles of SIRT2 in breast cancer, particularly in the initiation processes of breast cancer, such as DNA damage, genetic instability, and mitotic catastrophe, as this could provide insights into the early detection and treatment of breast cancers.

### 2.4. Cytoplasmic Sirtuins in Prostate Cancer

#### SIRT2

Unlike in breast cancer, SIRT2 was previously shown to be highly expressed in prostate cancer cells compared with normal prostate cells [86]. However, in a recent study, SIRT2 expression was shown to decline from benign to malignant and then metastatic prostate cancer [142]. To corroborate this finding, Lee et al. showed that SIRT2 protein levels are reduced in castrate-resistant prostate cancer. These findings suggest a tumor stage-dependent expression of SIRT2 in prostate cancer, such that the SIRT2 protein level reduces as the prostate tumor progresses. Unfortunately, to date, there are no studies that have investigated the mechanistic role of SIRT2 in prostate cancer. In this regard, future studies are required to determine the oncogenic or tumor-suppressing effect of SIRT2 in prostate cancer.

### 2.5. Mitochondrial Sirtuins in Breast Cancer

#### 2.5.1. SIRT3

The primary mitochondrial-localized deacetylase SIRT3 is known to play a vital role in mitochondrial metabolism [143]. Loss of SIRT3 culminates in hyperacetylation of mitochondrial proteins, which subsequently leads to the reduction of mitochondrial ability to produce ATP and induction of oxidative stress [144]. By deacetylating several enzymes involved in mitochondrial metabolism, such as isocitrate dehydrogenase and manganese superoxide dismutase, SIRT3 protects against pathological conditions, including aging-associated pathophysiologies [145,146]. Furthermore, the dual roles of SIRT3 in various types of cancers have been extensively reviewed [147], and their role in breast cancer is discussed below.

The expression of SIRT3 is decreased in human breast cancers [148,149]. To substantiate the decreased expression of SIRT3 in breast cancer, further investigations revealed that the *SIRT3* gene is deleted in approximately 20 percent of human cancers and 40 percent of breast cancers [149]. Corroborating this, the *SIRT3* gene is deleted in breast cancer more than in other cancers [148]. Armed with these findings, several researchers have investigated the tumor-suppressive role of SIRT3 in breast cancer. Finley et al. demonstrated that SIRT3 suppresses the Warburg effect and proliferation of human breast cancers [149]. Mechanistically, the authors demonstrated that SIRT3 opposes the Warburg effect by destabilizing hypoxia-inducible factor 1α (HIF1α), the main facilitator of increased glycolysis and lactate production during hypoxic conditions [149]. The authors also revealed that SIRT3 overexpression suppresses the proliferation of CAMA1 breast cancer cells in the presence of high glucose [149]. To corroborate their findings, the authors demonstrated that the knockout of SIRT3 promoted the Warburg effect in human breast cancer cells, thus promoting the survival of the breast cancer cells [149].

Additionally, Zou et al. demonstrated that the loss of SIRT3 (following treatment with SIRT3 shRNA) in MCF-7 breast cancer cells resulted in increased acetylation at lysine 413 of isocitrate dehydrogenase 2 (IDH2), a key enzyme in the Krebs cycle that oxidizes and decarboxylates isocitrate into α-ketoglutarate, and that SIRT3 loss decreases the level of IDH2 dimerization [67]. The authors further demonstrated that SIRT3 loss reduced IDH2 activities by decreasing IDH2 dimerization and that IDH2 acetylation at lysine 413 impairs mitochondrial respiration and detoxification, increases ROS production, and correlates with Luminal B breast cancer risk [67]. These findings suggest that SIRT3 expression may reverse the tumorigenic phenotype in breast cancer cells. Pinteric et al. demonstrated that *de novo* overexpression of SIRT3 downregulates the expression of vegfr1 (a proangiogenic protein involved in cell proliferation), EMT markers (vimentin and slug), lactate dehydrogenase A (LDHA; a glycolytic marker), peroxisome proliferator-activated receptor gamma coactivator 1-alpha (PGC1α), SIRT1, superoxide dismutase 2 (SOD2), and catalase (CAT) in MCF-7 breast cancer cells, thus indicating a tumor suppressor effect of SIRT3 in breast cancer [68]. Supporting this finding, a recent study demonstrated that the overexpression of SIRT3 promoted apoptosis and inhibited the proliferation of MCF-7 and MDA-MB-231 breast cancer cells [71].

Intriguingly, another recent study showed that while SIRT3 expression improved mitochondrial mass and potential metabolism (Warburg effect) and increased SOD2 and CAT expression, it also increased mitochondrial ROS, DNA damage, and expression of tissue inhibitor of metalloproteinase 1 (a major inhibitor of MMP9), which have been shown to promote malignant progression and metastasis [150]. Further, SIRT3 expression also induced the formation of multinucleated cells and apoptosis and inhibited the proliferation of MDA-MB-231 breast cancer cells [69].

Further supporting the tumor suppressor role of SIRT3 in breast cancer, SIRT3 suppresses estrogen-induced cytosolic and mitochondrial ROS production, diminishes estrogen-induced DNA synthesis, and upregulates p53 expression in MCF-7 breast cancer cells [70], despite its increasing antioxidant activities and cytosolic ROS. From the above-mentioned studies, it is evident that there is a well-established tumor suppressor role of SIRT3 in breast cancer.

To date, only one study has shown a possible tumor-promoting effect of SIRT3 in breast cancer. SIRT3 knockdown in MCF-7 cells by siRNA increased ROS production (196%), formation of autophagic bodies (47%), and apoptosis (23%); it also decreased MnSOD (20%), IDH2 (50%), PGC1α (12%), and mitochondrial transcription factor A (27%) levels and reduced cell viability (20%) [66]. These findings indicate that SIRT3 expression prevents ROS production, autophagy, and apoptosis, as well as increases cell viability and survival of breast cancer cells, thus demonstrating a tumor-promoting effect. However, given the limited information in this research area, further investigations are required to fully establish the tumor-promoting effect of SIRT3 before a conclusion can be reached.

#### 2.5.2. SIRT4

Like other mitochondrial sirtuins, SIRT4 is a mitochondrial deacetylase involved in energy metabolism [151]. However, in addition to its deacetylase activity, SIRT4 possesses ADP-ribosyl transferase activity, which catalyzes the transfer of ADP-ribosyl units to targets, such as glutamate dehydrogenase [9]. SIRT4 has been reported to be a therapeutic target that has both oncogenic and tumor suppressor effects in cancers [152]; however, the underlying mechanism is still a matter of debate. So far, most studies have demonstrated that SIRT4 regulates glutamine metabolism [153]. Notably, it is through the inhibition of glutamine metabolism that SIRT4 exhibits tumor suppressor effects in B Cell Lymphoma [154], 2014), colorectal cancer [155], and thyroid cancer [156].

In breast cancer, the expression pattern of SIRT4 is controversial. While some studies report that SIRT4 expression is upregulated in breast cancer [157,158], others report downregulation [97,129,153,159]. The differences in the expression patterns of SIRT4 in breast cancer cells may be attributed to differences in the breast cancer cell types and quantitation methods for SIRT4 used in these studies. Given its differential pattern of expression, it can be speculated that SIRT4 exhibits both tumor-suppressing and tumor-promoting effects. For instance, although the underlying mechanism was not investigated, a 2017 study demonstrated that SIRT4 upregulation promotes the proliferation, migration, and invasion of MDA-MB-435S breast cancer cells [158], although it should be noted that there is also controversy over whether this cell line is of breast origin [160]. Additionally, a recent study demonstrated that SIRT4 deacetylates methylenetetrahydrofolate dehydrogenase/methylenetetrahydrofolate cyclohydrolase 2 (MTHFD2), a vital bifunctional enzyme in folate metabolism, to remodel folate metabolism against breast cancer cells. As a mechanism, the authors revealed that SIRT4 deacetylation of MTHFD2 destabilizes MTHFD2 and causes its proteasomal degradation, leading to NADPH reduction and intracellular ROS accumulation in the breast tumors which, in turn, inhibits the proliferation of the breast cancer cells [73].

On the contrary, another study demonstrated that SIRT4 overexpression downregulates IL-6 expression and STAT3 Y705 phosphorylation, as well as the transcription and translation of STAT3 target genes (*MYC* and *CNDD1*), to enhance the sensitivity of MCF7 and T47D breast cancer cells to tamoxifen, thus potentiating the effect of this chemotherapy against breast cancer cells [72]. Supporting this, the downregulation of SIRT4 in breast cancer cell lines resulted in decreased expression of SIRT1 and stem cell markers Oct4, Sox2, and Nanog. Further, the SIRT4-mediated downregulation of SIRT1 led to increased acetylation of H4K16, which caused the downregulation of *BRCA1* expression at both mRNA and protein levels [74]. The authors suggested the crosstalk between the mitochondrial and nuclear sirtuins following SIRT1 downregulation by SIRT4 expression culminated in the suppression of breast cancer. This needs to be investigated further, considering that SIRT1 is constitutively upregulated in breast cancer and its inhibition suppresses breast cancer (as discussed in 2.1 above).

#### 2.5.3. SIRT5

In addition to its deacetylase activity, SIRT5 has been shown to exhibit demalonylase, deglutarylase, and succinylase activities [14,161]. As a member of the mitochondrial sirtuins, SIRT5, through its deacetylase activities, participates in several biological processes, such as metabolic regulation, aging, and oxidative stress [162]. SIRT5 expression has been evaluated in various kinds of cancer, particularly in head and neck squamous cell carcinoma [163], endometrial carcinoma [164], basal carcinoma [165], and hepatocellular carcinoma [166]. Although SIRT5 expression was found to be upregulated in breast cancer [76,159], which may be indicative of its role in breast cancer progression, only a few studies have revealed the mechanisms underlying the role of SIRT5 in breast cancer.

Polletta et al. demonstrated that ammonia production decreased in SIRT5 overexpressing MDA-MB-231 breast cancer cells, which was corroborated by the increase in ammonia production in SIRT5-silenced cells or cells treated with a specific inhibitor of SIRT5 (MC3482) [75]. The authors further demonstrated that SIRT5 inhibition leads to increased succinylation of glutaminase (GLS), an enzyme that catalyzes the conversion of glutamine to glutamate, and that ammonia increases the levels of autophagy markers (MAP1LC3B, GABARAP, and GABARAPL2) and mitophagy markers (BNIP3 and PINK1-PARK2 system). Taken together, the authors concluded that SIRT5 expression regulates ammonia-induced autophagy and mitophagy of MDA-MB-231 breast cancer cells by inhibiting GLS activity. On the contrary, a recent study demonstrated that SIRT5 desuccinylates GLS at residue K164, stabilizing and protecting it from ubiquitin-mediated degradation and that SIRT5 expression promotes the proliferation and tumorigenesis of MDA-MB-231 breast cancer cells [76]. Given the disagreement between these two studies regarding the effect of SIRT5 on GLS activities, further clarification by future studies is required before a conclusion can be reached.

Importantly, an in vivo study of SIRT5 knockout MMTV-PyMT mice supported the tumor-promoting effect of SIRT5 in breast cancer. SIRT5 knockout MMTV-PyMT mice had higher survival and reduced tumor size and lung metastases compared to the SIRT5 wild-type MMTV-PyMT mice [77]. The authors further demonstrated that SIRT5 knockout cancer cells had increased levels of ROS and reduced levels of important antioxidants NADPH and GSH. As such, the authors concluded that SIRT5 may promote breast cancer by mitigating ROS via one or several of its targets [77]. To date, most studies suggest that SIRT5 acts as a tumor promoter in breast cancer. Consistent with this deduction, a recent study established SIRT5 inhibition as a crucial promising therapeutic approach against breast cancer in vivo and in vitro [167].

### 2.6. Mitochondrial Sirtuins in Prostate Cancer

#### 2.6.1. SIRT3

The expression of SIRT3 in prostate cancer remains controversial. SIRT3 expression has been reported to be decreased in prostate cancer by two studies: Quan et al. [92] found that SIRT3 is moderately downregulated in prostate carcinoma tissues and Li et al. [93] detected a decrease in metastatic tissues compared to prostate tumor tissues. Singh et al. [94] found that SIRT3 was upregulated in cancerous prostatic tissues and cell lines compared to normal tissues and prostatic epithelia cells, respectively. These conflicting findings may suggest a differential pattern of SIRT3 expression in prostate cancer tissues and cells, which may depend on the cell type, tumor stage or grade, and genetic differences in the prostate cancer tissues or cells. In this regard, further studies are needed to clarify the expression pattern of SIRT3 in prostate cancer tissues and cells. Similarly, mechanistic studies have been conducted in prostate cancer, which indicated that SIRT3 has a dual role in prostate carcinogenesis.

Supporting a tumor suppressive function, SIRT3 expression in C42B and PC3 prostate cancer cells downregulated the expression of c-Myc [92], a well-established oncogenic protein [168]. The same authors further demonstrated that SIRT3 expression destabilizes c-Myc by inhibiting the PI3k/Akt pathway (in vitro and in vivo) and that SIRT3 expression suppresses the PI3K/Akt pathway by downregulating ROS levels. Therefore, the authors concluded that SIRT3 expression inhibits the proliferation of prostate cancer via the above-mentioned mechanisms. Corroborating this finding, Li et al. found that SIRT3 expression suppressed EMT (via upregulation of E-cadherin and downregulation of N-cadherin and vimentin expression) in C42B prostate cancer cells by upregulating FOXO3A [93], a transcription factor that has been shown to suppress EMT and metastasis of prostate cancer [169]. They also found that SIRT3 overexpression downregulates the expression of β-catenin in C42B prostate cancer cells, thereby suppressing the Wnt/β-catenin signaling, a pathway that has been shown to inhibit the nuclear localization of FOXO3A and the expression of *FOXO3A* target genes [170]. Put together, the authors concluded that SIRT3 inhibits the metastasis of prostate cancer via the Wnt/β-catenin/FOXO3A signaling axis. Consistent with this tumor suppressing effect of SIRT3, a recent study demonstrated that SIRT3 expression reduces the acetylation level of mitochondrial aconitase, leading to the reduction of prostate cancer growth [95].

In contrast with the above-mentioned studies indicating the tumor suppressor function of SIRT3 in prostate cancer, Singh et al. demonstrated that SIRT3 expression inhibits the cleavage of poly (ADP-ribose) polymerase [94], a known marker of apoptosis, and upregulates the expression of proliferating cell nuclear antigen, a known marker of cell proliferation, in DU145 and 22Rν1 prostate cancer cells. The authors concluded that SIRT3 has a pro-proliferative role in prostate cancers. Furthermore, a recent study revealed that SIRT3 suppresses necroptosis-induced innate immunity to promote the progression of prostate cancer [96]. To validate this finding, the authors demonstrated that the knockdown of SIRT3 led to the recruitment of macrophages and neutrophils and induced necroptosis in prostate cancer cells. Notwithstanding, future mechanistic studies should be channeled toward validating these findings.

#### 2.6.2. SIRT4

Unlike in other human cancers, including breast cancer, the expression pattern of SIRT4 and its mechanistic roles in prostate cancer have been sparsely investigated. Nevertheless, a study found that SIRT4 suppresses the proliferation of DU145 prostate cancer cells by inhibiting the uptake of glutamine [97], a metabolite vital for proliferating and cancerous cells [171]. Furthermore, Li et al. demonstrated that SIRT4 promotes mitochondrial-mediated apoptosis in PC3 prostate cancer cells by deacetylating and promoting the ubiquitination and degradation of adenine nucleotide translocase-2 [98], a mitochondrial inner membrane protein that is highly expressed in cancer cells [172]. Notwithstanding, further studies are required to clarify the expression pattern of SIRT4 and its mechanistic roles in prostate cancer.

#### 2.6.3. SIRT5

As in breast cancer, SIRT5 was found to be highly expressed in prostate tumor tissues compared to normal prostate tissues [99]. However, in contrast, recent studies have revealed that SIRT5 expression is downregulated in advanced prostate cancer [100,101], suggesting that SIRT5 downregulation is dependent on the progression or stage of the prostate cancer cells. These findings may indicate a tumor-stage-dependent role of SIRT5 in prostate carcinogenesis. There are few studies on the mechanistic roles of SIRT5 in prostate cancer. A recent study demonstrated that SIRT5 promotes the proliferation and migration of LNCaP and PC3 prostate cancer cells [99]. This was shown to occur through increased expression of SIRT5, which resulted in the upregulation of cyclin D1, MMP9, and MAPK signaling-related proteins. The authors further demonstrated that SIRT5 decreased the protein levels of acetyl-CoA acetyltransferase 1 (ACAT1), a promoter of prostate cancer [173]. Since ACAT1 negatively regulates the MAPK signaling pathway [99], Peck and Shulze [173] concluded that SIRT5 promotes the proliferation, invasion, and migration of prostate cancer via the downregulation of ACAT1 protein levels.

Conversely, a more recent study demonstrated that SIRT5 expression reduces the migration and invasion of PC3 prostate cancer cells [101]. This was found to occur via SIRT5 desuccinylation of LDHA to reduce its activity, culminating in the growth reduction and survival of prostate cancer cells [101]. Consistent with this finding, another recent study demonstrated that SIRT5 suppressed the progression of PC3 prostate cancer cells by downregulating the PI3K/AKT/NK-kB pathway [100]. Interestingly, the authors further revealed that SIRT5-facilitated downregulation of the PI3K/AKT/NK-kB pathway reduced as the prostate cancer metastasized from the bone to other tissues [100]. The findings of these two recent studies indicate that the tumor-suppressing effect of SIRT5 in prostate cancer increases as the prostate cancer progresses but reduces at the point of secondary metastasis (from the bone to other tissues). Nevertheless, further studies are required to corroborate the current findings and investigate the consequences of the modulation of other targets by SIRT5 in prostate cancer.

## 3. Mechanistic Role of Sirtuins Regulation by microRNAs (miRNAs) in Breast and Prostate Cancers

### 3.1. Overview of miRNA

In 1993, the first miRNA *lin-4* was reported to exist in *Caenorhabditis elegans* [174,175]. After a series of intensive scientific investigations, the same groups found that *lin-4* was not an RNA-encoded protein but a non-coding RNA [176,177]. Furthermore, they found that *lin-4* post-transcriptionally downregulates *lin-14* (a gene found in *Caenorhabditis elegans*) through its 3′ untranslated region (UTR) and that the 3′UTR of *lin-14* was complementary to that of *lin-4* [175]. Accordingly, they proposed that *lin-4* post-transcriptionally regulates *lin-14* [174]. Following this landmark discovery, over 1,000 miRNAs have been detected in various species [178].

miRNAs are small non-coding RNAs composed of an average of 22 nucleotides that suppress gene expression by interacting with the 3′ UTR of target mRNAs. Moreover, several studies have demonstrated that miRNAs repress mRNA translation and induce mRNA deadenylation and decapping by binding to the 3′ UTR of their target mRNAs [179,180]. In addition, other regions in the target mRNA, such as the promoter region, 5′ UTR, and coding region have been shown to have miRNA binding sites [181]. According to scientific findings, when miRNAs bind to the 5′ UTR and coding region of their target mRNA, they induce the silencing of gene expression [182]; whereas, when they interact with the promoter region of their target mRNA, they induce gene transcription [183].

miRNAs have been implicated in normal biological processes [184,185,186], as well as in various human diseases [187,188]. Based on this fact, several studies have explored the diagnostic and prognostic potential of miRNAs in various human diseases [189,190,191,192]. Furthermore, in various cancer diseases, miRNAs have been demonstrated to confer either oncogenic or tumor suppressor effects [193,194,195,196]. Most importantly, different roles of miRNA, including tumorigenesis, progression, metastasis, diagnosis, and prognosis, in breast [197,198,199,200] and prostate [201,202,203,204] cancer have been reported in the literature. The current knowledge on the regulation of SIRT1–7 by miRNAs will be discussed in a subsequent section of this review, highlighting the consequences of this regulation in breast and prostate cancer.

Reportedly, approximately 30% of genes are regulated by miRNAs. Consistent with this report, the regulatory functions of miRNAs in the progression, metastasis, and drug-resistance of breast tumors and other cancers have been documented [198,205]. Although miRNAs are well-established as regulators of various proteins involved in signaling pathways in breast and prostate cancer, there is ongoing research on the regulation of sirtuins by miRNAs in breast and prostate cancer. Given the limited information in the literature to date, we discuss the current information on the mechanistic role of miRNA-mediated regulation of SIRT1 and 7 in breast cancer (Table 3) and SIRT1 in prostate cancer (Table 4) and elucidate a common mechanism for this regulation (Figure 4).

### 3.2. SIRT1 Regulation by miRNAs in Breast Cancer

Several miRNAs have been found to regulate SIRT1 expression in breast cancer, serving as either tumor suppressors or tumor promoters through an indirect mechanism, depending on their effect on SIRT1 expression. For instance, Eades et al. demonstrated that miR-200a epigenetic silencing contributes, at least in part, to the overexpression of SIRT1, and the SIRT1 transcript is subject to regulation by miR-200a via SIRT1 3′-UTR [211]. They further demonstrated that miR-200a expression or SIRT1 knockdown prevents the transformation of normal mammary epithelial cells and that overexpression of SIRT1 is associated with decreased miR-200a in breast cancer patients. Furthermore, Li et al. (2013b) found that miR-34a is downregulated in BT474, MDA-MB-231, MDA-MB-435 (although this may not be relevant due to a postulation that it is a melanoma cell line), MDA-MB-468, and SK-BR-3 breast cancer cells. The same investigators also found that miR-34a expression (ectopic) suppressed the proliferation, invasion, and induced apoptosis of these breast cancer cells. They further demonstrated that Bcl-2 and SIRT1 are targets of miR-34a and are negatively correlated with the ectopic expression of miR-34a. Moreover, miR-34a has been well-established as a tumor suppressor [220,221].

Furthermore, a study demonstrated that miR-22 is downregulated in MCF-7 and MDA-MB-231 breast cancer cells, and the overexpression of miR-22 or SIRT1 knockdown induces apoptosis and decreases the survival of breast cancer cells [209]. The authors further demonstrated that miR-22 negatively regulates the expression of SIRT1, and SIRT1 ectopic expression reversed the inductive effect of miR-22 on apoptosis and the radiosensitivity of breast cancer cells. This indicates a tumor suppressor role of miR-22 through the negative regulation of SIRT1. Similarly, Liang et al. reported that miR-4766-5p negatively regulates SIRT1 expression to suppress cell proliferation, metastasis, and chemoresistance in breast cancer cells [208]. Another study found that miR-590-3p expression is downregulated in MCF-7 and MDA-MB-231 breast cancer cells, and miR-590-3p exerts its antitumor effects (induction of apoptosis and reduction of cell survival) by targeting SIRT1 in breast cancer cells [207]. Another study demonstrated that SIRT1 expression is inhibited by miR-211-5p by targeting SIRT1 via its 3′-UTR, and miR-211-5p expression culminated in decreased acetylation of p53, which was associated with reduced cell viability and apoptosis induction in breast cancer cells [206]. The authors concluded that the negative regulation of SIRT1 by miR-211-5p reduces survival and induces cell death of breast cancer. So far, the information provided above reveals that most miRNAs targeting SIRT1 act as tumor suppressors in breast cancer; however, studies are needed to clarify and verify the possible tumor-promoting effects of SIRT1 negative regulation by miRNAs in breast cancer. Additionally, future studies should also attempt to match the effects of the miRNA with the SIRT1 functions in breast cancer discussed above.

### 3.3. SIRT7 Regulation by miRNAs in Breast Cancer

Based on our literature search, there is limited information on the regulation of SIRT7 by miRNAs in breast cancer. Being the only available data at the time of our literature search, Li and Li [212] demonstrated that miR-3666 expression is reduced in MCF-7, BT474, MDA-MB-231, and MDA-MB-468 breast cancer cells and that miR-3666 overexpression inhibits the proliferation of breast cancer cells. The authors further demonstrated that miR-3666 targets the SIRT7 3′-UTR, and miR-3666 overexpression reduces the expression of SIRT7, resulting in increased proliferation and reduced apoptosis of breast cancer cells [212]. The reduced miR-3666 expression and increased SIRT7 expression in MDA-MB-231 breast cancer cells, as mentioned above, suggest their prognostic and diagnostic relevance in breast cancer. Surprisingly, miR-3666 was recently established as a tumor suppressor in different cancers, such as head and neck squamous cell carcinoma (HNSCC) [222] and glioblastomas (GMB) [223]. However, it is worth noting that SIRT7 expression is downregulated in human HNSCC tissues [163] but upregulated in human glioma tissues [224], and >50% of gliomas are reported to be GMB [225]. Therefore, it may be hypothesized that a reduced expression of miR-3666 is corroborated by an increased expression of SIRT7 in GMB but not in HNSCC. Given these scenarios, further investigations are required to corroborate the present findings, ascertain the tumor-promoting roles of miRNAs via regulation of SIRT7 in breast cancer, as well as determine how miRNAs-induced effects match with the SIRT7 functions in breast cancer discussed above.

### 3.4. SIRT1 Regulation by miRNAs in Prostate Cancer

Although several studies have investigated the role of miRNAs, including miR-1, -21, -106b, -141, -145, -182, -187, -205, -221, -375, -490-3p, and -Let-7c in the prognosis, diagnosis, and progression of prostate cancer [226,227,228,229], there is limited information on the modulation of sirtuins by miRNAs in prostate cancer. Given the tumor-promoting role of SIRT1 in prostate cancer, which is evident in its high expression in prostate cancer [87], several studies have investigated the modulatory roles of various miRNAs on SIRT1 in prostate cancer, suggesting a tumor suppressor role of miRNAs through the inhibition of SIRT1 expression. For instance, Fujita et al. (2008) demonstrated that miR-34a expression is downregulated in PC3 and DU145 prostate cancer cells, and ectopic expression of miR-34a downregulates SIRT1 protein expression in PC3 prostate cancer cells at the transcriptional level (that is, via reduction of SIRT1 promoter activity) but not at the post-transcriptional level (that is, via interaction with SIRT1 3′-UTR). The authors further demonstrated that the ectopic expression of miR-34a induces cell cycle arrest, inhibits cell growth, and abrogates camptothecin resistance of PC3 prostate cancer cells. Taken together, these findings indicate a tumor suppressor role of miR-34a through the negative regulation of SIRT1 in prostate cancer. Moreover, a subsequent study confirmed that miR-34a expression is downregulated in prostate cancer tissues compared to normal prostate tissues, and miR-34a expression inhibits the proliferation of PC3 prostate cancer cells by downregulating SIRT1 expression [219].

miR-212 was found to be downregulated in prostate cancer tissues compared with normal prostate tissues [218]. Its expression inhibits SIRT1 expression (via interaction with SIRT1 3’UTR), starvation- and SIRT1-induced autophagy, and angiogenesis of LNCaP and PC3 prostate cancer cells [218]. These findings suggest a tumor suppressor role of miR-212 through the negative regulation of SIRT1 in prostate cancer. Other subsequent studies have also shown that certain miRNAs, whose expressions are downregulated in prostate cancer, act as tumor suppressors by negatively regulating SIRT1 via its 3′UTR. For instance, by inhibiting SIRT1 protein expression, researchers demonstrated that miR-449a ectopic expression suppresses the invasiveness of prostate cancer [217]; miR-204 enhances both docetaxel-induced apoptosis [214] and doxorubicin-induced mitochondrial-mediated apoptosis [216] of prostate cancer, and miR-138-5p inhibits the proliferation and lipid metabolism of prostate cancer [215].

So far, most of the above-mentioned studies suggest that miRNAs directly interact with SIRT1 to potentiate their tumor-suppressing effect in prostate cancer. On the contrary, Yang et al. demonstrated that miR-221 and miR-222 are highly expressed in PC3 compared to LNCaP prostate cancer cells and do not target or directly interact with SIRT1 mRNA via its 3′UTR [230]. Interestingly, the authors further showed that the inhibition of these miRNAs reduces proliferation, increases apoptosis, and reduces migration of PC3 prostate cancer cells, which were abrogated by SIRT1 expression. This suggests interesting crosstalks between these miRNAs and SIRT1 in prostate cancer, which needs to be explored in future studies. As in breast cancer, there is currently no study demonstrating the tumor-promoting effect of miRNAs via the SIRT1 axis in prostate cancer.

## 4. Conclusions

This extensive review of the current knowledge on the mechanistic roles of nuclear sirtuins (SIRT1, 6, and 7), cytoplasmic sirtuins (SIRT2), and mitochondrial sirtuins (SIRT3, 4, and 5) has shown that breast and prostate cancer cells or tissues portray different expression patterns of the various types of sirtuins to aid their tumorigenic phenotypes and potentiate their progression through various carcinogenic stages. Furthermore, we have also shown that sirtuins regulate various proteins to promote or suppress breast and prostate cancer. Specifically, we highlighted that sirtuins regulate various proteins implicated in proliferation, apoptosis, autophagy, chemoresistance, invasion, migration, and metastasis of both breast and prostate cancer. To further illustrate the relevance of sirtuins expression in breast and prostate cancer, we have provided available evidence of the direct regulation of sirtuins by miRNAs (via direct interaction with sirtuins mRNAs), highlighting the consequences of this regulation in breast and prostate cancer. While many studies have begun to investigate the specific mechanisms by which sirtuins influence either breast or prostate cancer, no study has investigated both in the same study. As a result, the specific investigation of either of these cancers has focused on different key genes and/or proteins involved in various molecular pathways. Comparing Figure 3a and Figure 3b, it is evident that similar functions of sirtuins are relied upon by both cancers; however, the specific genes driving these functions may be different. Future studies verifying these genetic mechanisms in both cancers will be imperative to determine if sirtuins can be targeted in the same way as novel treatments for these cancers. In general, this review highlights that sirtuins show similar differences in expression in both breast and prostate cancer, which then drive similar molecular mechanisms to promote the progression of cancer and thus represent new therapeutic targets. Notwithstanding, research on the mechanistic role of sirtuins and consequences of their regulation by miRNAs in breast and prostate carcinogenesis is ongoing. Future studies are required to illuminate the grey areas and validate the current knowledge. Nonetheless, this review has highlighted the potential value of sirtuins as biomarkers and/or targets for improved treatment of breast and prostate cancers.

## Figures and Tables

**Figure 1 cancers-14-05118-f001:**
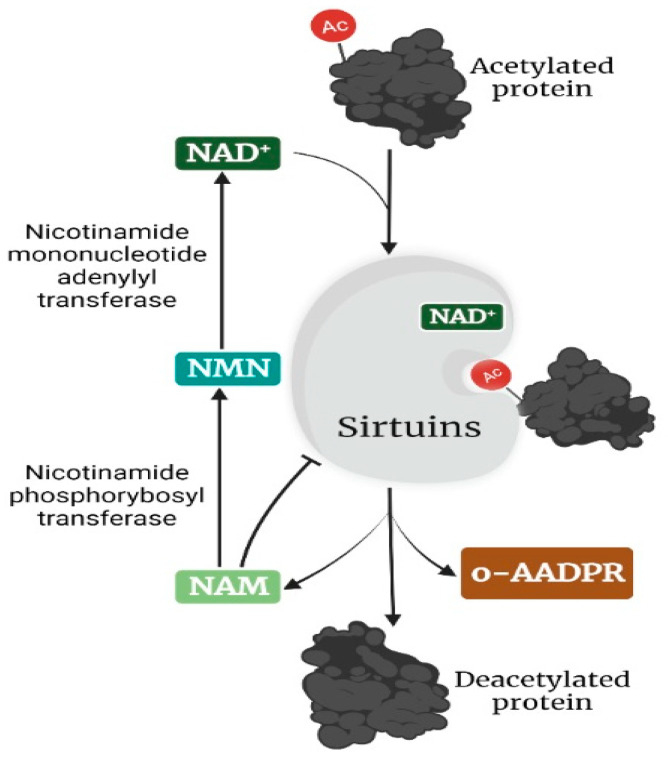
Mechanism of deacetylation of targets by sirtuins. In the presence of the co-factor NAD^+^, the acetylated protein target is deacetylated by sirtuins, producing a deacetylated protein, o-AADPR, and NAM, a feedback inhibitor of sirtuins. NAM is reconverted through NMN to NAD^+^ in two enzyme-catalyzed steps. The image was created with BioRender.com.

**Figure 2 cancers-14-05118-f002:**
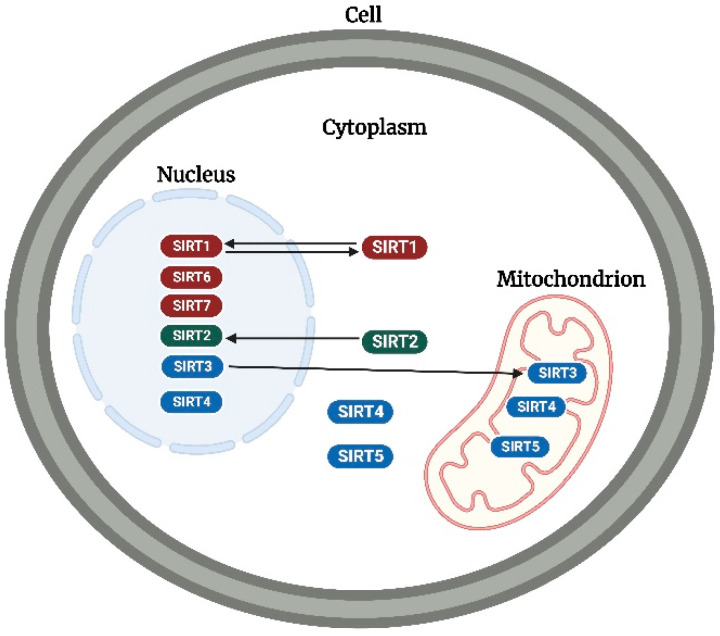
Subcellular location of sirtuins. SIRT1, 6, and 7 (in red), SIRT2 (in green), and SIRT3, 4, and 5 (in blue) are predominantly located in the nucleus, cytoplasm, and mitochondrion, respectively. SIRT1, 2, and 3 can translocate to the cytoplasm, nucleus, and mitochondrion, respectively. While SIRT 3 and 5 have been detected in the nucleus and cytoplasm, respectively, SIRT4 has been detected in both the cytoplasm and nucleus. The image was created with BioRender.com.

**Figure 3 cancers-14-05118-f003:**
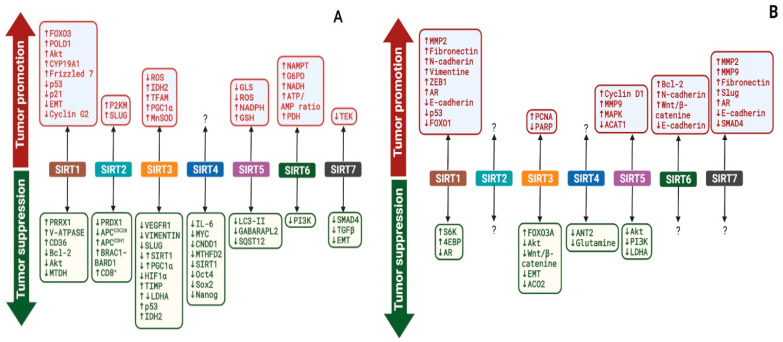
Outcomes of sirtuins modulation of various targets in breast (**A**) and prostate (**B**) cancers. SIRT1–7 regulate the expression and activities of various targets in breast (**A**) and prostate (**B**) cancer, resulting in either tumor promotion or suppression. ?—Not Known. The image was created with BioRender.com.

**Figure 4 cancers-14-05118-f004:**
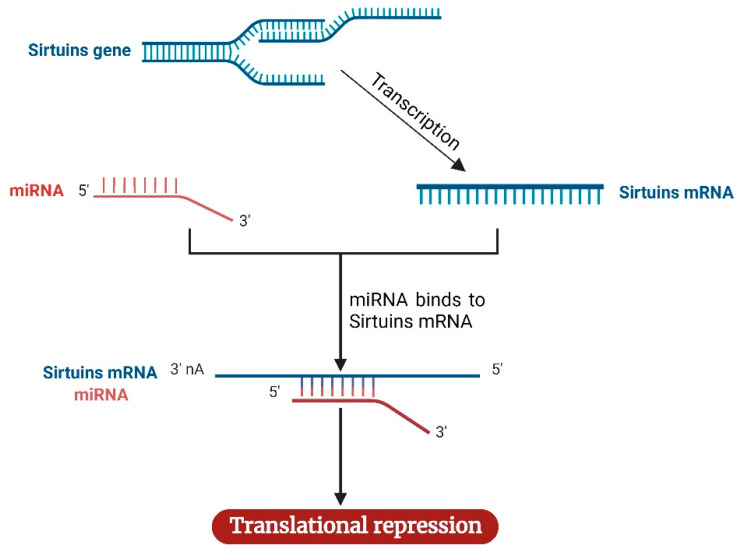
Mechanism of sirtuins regulation by miRNAs. Following the transcription of the sirtuins gene to mRNA, miRNA complementary to sirtuins binds the sirtuins mRNA, causing the repression of sirtuins mRNA translation. The image was created with BioRender.com.

**Table 1 cancers-14-05118-t001:** Mechanistic Roles of Sirtuins (SIRT1–7) in Breast Cancer.

Sirtuins	Mechanism/Target	Function	References
SIRT1	Deacetylates and stabilizes PRRX1	Inhibits breast cancer stemness and metastasis	[46]
SIRT1	Upregulates V-ATPase expression and activity	Inhibits secretion of exosomes, which cause aggressiveness of breast cancer cells	[47]
SIRT1	Upregulates constitutive high FOXO3 expression	Promotes drug (lapatinib) resistance of breast cancer	[48]
SIRT1	Upregulates DNA POLD1	Promotes proliferation, migration, and invasion of breast cancer	[49]
SIRT1	Upregulates Akt activity	Promotes formation of breast cancer	[50]
SIRT1	Downregulates Bcl-2 protein	Suppresses growth of breast cancer	[51] *
SIRT1	Upregulates CYP19A1 expression	Promotes growth of estrogen-dependent breast cancer	[52]
SIRT1	Prevents p53 activation and increased p21 expression	Promotes growth of breast cancer	[53]
SIRT1	Upregulates expression of Frizzled 7	Promotes proliferation of breast cancer	[54]
SIRT1	Represses ERα activation via Akt deactivation	Suppresses growth of estrogen-dependent breast cancer	[55]
SIRT1	Upregulates EMT-related proteins	Promotes invasion and metastasis of breast cancer	[56]
SIRT1	Prevents c-Myc-mediated upregulation of MTDH	Inhibits proliferation of breast cancer	[57]
SIRT1	Upregulates expression of CD36	Inhibits proliferation of breast cancer	[58]
SIRT1	Inactivates cyclin G2 and p53	Promotes growth of estrogen-dependent breast cancer	[59]
SIRT2	Deacetylates and inhibits the peroxidase activity of peroxiredoxin-1	Suppresses growth of breast cancer	[60]
SIRT2	Deacetylates K116 in the SLUG domain to stabilize slug	Promotes growth and aggressiveness of basal-like breast cancer	[61]
SIRT2	Deacetylates APC^CDH1^ and APC^CDC2^	Suppresses growth of breast cancer	[62]
SIRT2	Deacetylates and activates pyruvate kinase (PKM2)	Promotes growth of breast cancer	[63]
SIRT2	Deacetylates K116 in the SLUG domain to stabilize slug	Promotes growth and aggressiveness of basal-like breast cancer	[61]
SIRT2	Deacetylates APC^CDH1^ and APC^CDC2^	Suppresses growth of breast cancer	[62]
SIRT2	Promotes BRCA1-BARD1 heterodimerization via deacetylation	Suppresses growth of breast cancer	[64]
SIRT2	Regulates CD8+ effector memory T-cells differentiation	Induces immune response against breast cancer	[65]
SIRT3	Downregulates ROS production and upregulates MnSOD, IDH2, PGC1-α, and TFAM	Increases cell viability and inhibits autophagy and apoptosis of breast cancer	[66]
SIRT3	Deacetylates and increases IDH2 activities	Suppresses the growth of breast cancer	[67]
SIRT3	Downregulates expression of the angiogenic gene *vegfr1*, EMT markers (*vimentin* and *slug*), LDHA, antioxidant genes (*sod2* and *cat*), SIRT1 and PGC1α	Inhibits survival, proliferation, and mitochondrial function in breast cancer	[68]
SIRT3	Upregulates expression of SIRT1, LDHA, and PGC-1α; increases mitochondrial ROS, induces DNA damage, timp-1 expression, formation of multinucleated cells, and apoptosis.	Improves mitochondrial function but inhibits proliferation of breast cancer	[69]
SIRT3	Upregulates p53 and attenuates the response to estrogen	Suppresses growth of estrogen-dependent breast cancer	[70]
SIRT3	Participates in PGC-1α suppression of glycolytic metabolism	Inhibits proliferation of breast cancer	[71]
SIRT4	Downregulates IL-6, STAT3 Y705 phosphorylation as well as transcription and translation of STAT3 target genes (*MYC* and *CNDD1*)	Enhances the sensitivity of breast cancer cells to tamoxifen	[72]
SIRT4	Deacetylates MTHFD2 at K50	Inhibits growth of breast cancer	[73]
SIRT4	Downregulates SIRT1 expression and stem cell markers (Oct4, Sox2, and Nanog)	Suppresses the self-renewal of breast cancer stem cells	[74]
SIRT5	Regulates glutamine metabolism, suppresses LC3-II and GABARAPL2 accumulation as well as sequestosome 1 degradation	Induces mitophagy and autophagy of breast cancer	[75]
SIRT5	Desuccinylates and stabilizes mitochondrial enzyme GLS	Promotes breast cancer tumorigenesis	[76]
SIRT5	Reduces ROS generation and increases NADPH and GSH levels	Promotes tumor progression and metastasis of breast cancer	[77]
SIRT6	Deacetylates and activates NAMPT and glucose-6-phosphate dehydrogenase activities and increases NADH levels	Promotes breast cancer survival and resistance to oxidative stress	[78]
SIRT6	Downregulates PI3K signaling at the transcriptional level independent of its deacetylase activity	Suppresses progression and stemness of breast cancer	[79]
SIRT6	Enhances pyruvate dehydrogenase expression and activity, oxidative phosphorylation, and ATP/AMP ratio	Promotes growth of breast cancer	[80]
SIRT7	Deacetylates and promotes SMAD4 degradation mediated by β-TrCP1; downregulates TGFβ and prevents epithelial-to-mesenchymal transition	Inhibits metastasis of breast cancer	[81]
SIRT7	Deacetylates the TEK promoter at H3K18	Promotes Adriamycin-induced metastasis of breast cancer	[82]

* Title of study contradicts the study findings.

**Table 2 cancers-14-05118-t002:** Mechanistic Roles of Sirtuins (SIRT1–7) in Prostate Cancer.

Sirtuins	Mechanism/Target	Function	References
SIRT1	Deacetylates and upregulates Matrix Metalloproteinase-2 (MMP2) expression	Promotes prostate cancer cell invasion	[83]
SIRT1	Downregulates EMT-related protein (E-cadherin) and upregulates mesenchymal markers (vimentin and N-cadherin)	Promotes movement, migration, and invasion of prostate cancer cells	[84]
SIRT1	Downregulates expression of epithelial marker (E-cadherin) and upregulates expression of mesenchymal markers (N-cadherin and fibronectin) and EMT-inducing transcription factor (ZEB1)	Promotes migration and metastasis of prostate cancer	[85]
SIRT1	Deacetylates and deactivates p53 and FOXO1	Promotes development of prostate cancer	[86]
SIRT1	Upregulates phosphorylation of S6K and 4EBP1	Suppresses cell proliferation and induces autophagy of prostate cancer	[87]
SIRT1	Antagonizes PCAF-catalyzed MPP8-K439 acetylation	Promotes migration, invasion, and EMT of prostate cancer	[88]
SIRT1	Deacetylates AR and histone H3, and suppresses AR-mediated gene transcription	Inhibits proliferation of prostate cancer	[89]
SIRT1	Deacetylates AR and inhibits coactivator-induced interactions between AR amino and carboxyl termini	Inhibits proliferation of prostate cancer	[90]
SIRT1	Upregulates AR signaling	Promotes progression of prostate cancer	[91]
SIRT2	Not known	Not Known	Not Known
SIRT3	Inhibits phosphorylation of Akt, leading to ubiquitination and degradation of oncoprotein c-Myc	Inhibits proliferation of prostate cancer	[92]
SIRT3	Promotes FOXO3A expression by suppressing Wnt/β-catenin pathway; Downregulates EMT	Inhibits migration and metastasis of prostate cancer	[93]
SIRT2	Not known	Not Known	Not Known
SIRT3	Inhibits phosphorylation of Akt, leading to ubiquitination and degradation of oncoprotein c-Myc	Inhibits proliferation of prostate cancer	[92]
SIRT3	Promotes FOXO3A expression by suppressing Wnt/β-catenin pathway; Downregulates EMT	Inhibits migration and metastasis of prostate cancer	[93]
SIRT3	Inhibits cleavage of poly (ADP-ribose) polymerase (PARP) and upregulates expression of proliferating cell nuclear antigen	Promotes proliferation and suppresses apoptosis of prostate cancer	[94]
SIRT3	Reduces the level of acetylated mitochondrial ACO2	Reduces growth and survival of prostate cancer	[95]
SIRT3	Inhibits RIPK3-mediated necroptosis and innate immune response	Promotes progression of prostate cancer	[96]
SIRT4	Decreases glutamine uptake and metabolism	Inhibits proliferation of prostate cancer	[97]
SIRT4	Deacetylates ANT2 to promote its ubiquitination and degradation	Suppresses proliferation and promotes mitochondrial-mediated apoptosis of prostate cancer	[98]
SIRT5	Upregulates cyclin D1, MMP9, and MAPK signaling proteins; downregulates ACAT1 protein expression	Promotes proliferation, invasion, and migration of prostate cancer	[99]
SIRT5	Inhibits the PI3K/AKT signaling	Suppresses growth and metastasis of prostate cancer	[100]
SIRT5	Desuccinylates LDHA	Reduces migration and invasion of prostate cancer	[101]
SIRT6	Increases Bcl-2 gene expression and induces cell cycle arrest at the sub-G1 phase	Inhibits apoptosis and promotes survival and proliferation of prostate cancer	[102]
SIRT6	Downregulates E-cadherin level and upregulates N-cadherin level	Promotes migration and invasion of prostate cancer	[103]
SIRT6	Inhibits RIPK3-mediated necroptosis and innate immune response	Promotes progression of prostate cancer	[96]
SIRT6	Upregulates the Wnt/β-catenin signaling	Promotes progression of prostate cancer	[104]
SIRT7	Upregulates expression of EMT marker (fibronectin)	Promotes aggressiveness of prostate cancer	[105]
SIRT7	Upregulates AR signaling, LC3BI to LC3BII conversion, vimentin, slug, MMP2, and MMP9; downregulates SMAD4 protein expression	Promotes cell proliferation, metastasis, and androgen-induced autophagy of prostate cancer	[106]
SIRT7	Downregulates EMT-related protein (E-cadherin) and upregulates mesenchymal markers (vimentin) and slug	Promotes migration and metastasis of prostate cancer	[107]

**Table 3 cancers-14-05118-t003:** Mechanistic Roles of Sirtuins (SIRT1–7) Regulation by miRNAs in Breast Cancer.

Sirtuins	Regulatory microRNA	Function	References
SIRT1	miR-211-5p	Reduces cell viability and induces apoptosis of breast cancer	[206]
SIRT1	miR-590-3P	Induces apoptosis and reduces survival of breast cancer	[207]
SIRT1	miR-4766-5p	Suppresses cell proliferation, metastasis, and chemoresistance in breast cancer	[208]
SIRT1	miR-22	Suppresses tumorigenesis and improves radiosensitivity of breast cancer	[209]
SIRT1	miR-34a	Inhibits proliferation and migration of breast cancer	[210]
SIRT1	miR-200a	Prevents growth and EMT-like transformation in breast cancer	[211]
SIRT2	Not Known	Not Known	Not Known
SIRT3	Not Known	Not Known	Not Known
SIRT4	Not Known	Not Known	Not Known
SIRT5	Not Known	Not Known	Not Known
SIRT6	Not Known	Not Known	Not Known
SIRT7	miR-3666	Inhibits proliferation and promotes apoptosis of breast cancer	[212]

**Table 4 cancers-14-05118-t004:** Mechanistic Roles of Sirtuins (SIRT1–7) Regulation by miRNAs in Prostate Cancer.

Sirtuins	Regulatory microRNA	Function	References
SIRT1	miR-34a	Reduces growth and chemoresistance of prostate cancer	[213]
SIRT1	miR-204	Enhances docetaxel-induced apoptosis of prostate cancer	[214]
SIRT1	miR-138-5p	Inhibits proliferation and lipid metabolism of prostate cancer	[215]
SIRT1	miR-204	Enhances doxorubicin-induced mitochondrial-mediated apoptosis of prostate cancer	[216]
SIRT1	miR-449a	Suppresses invasiveness of prostate cancer	[217]
SIRT1	miR-212	Inhibits starvation- and SIRT1-induced autophagy of prostate cancer	[218]
SIRT1	miR-34a	Inhibits proliferation of prostate cancer	[219]
SIRT2	Not known	Not Known	Not Known
SIRT3	Not Known	Not Known	Not Known
SIRT4	Not Known	Not Known	Not Known
SIRT5	Not Known	Not Known	Not Known
SIRT6	Not Known	Not Known	Not Known
SIRT7	Not Known	Not Known	Not Known

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
