# Peer review of "The Mechanistic Roles of Sirtuins in Breast and Prostate Cancer"

_cancers, 2022, doi:10.3390/cancers14205118_

Round 1

Reviewer 1 Report

The authors have compiled and narrated the dual role of sirtuin genes and their proteins in the promotion and suppression of breast and prostate cancers. They discuss the reports that provide information on the altered expressions of different sirtuins in cancers of the breast and prostate. Furthermore, the expression of several proteins linked to various cellular processes that promote or suppress breast and prostate cancer that are regulated by sirtuins are described. Then, the authors present available limited information on the regulation of sirtuins by miRNAs to assess the consequences of miRNAs-mediated regulation of sirtuins in breast and prostate cancer.

A few suggestions and comments for improvements are as follows:

1. As with other classes of histone deacetylases (class I, II, and IV HDACs), sirtuins (class III HDACs) are involved in several cellular processes affecting different protein targets by deacetylase activity.

However, sirtuins are different from class I, II, and class IV in that some of the sirtuins have additional activities besides deacetylation. The authors have cited sirtuins’ function as NAD-dependent deacetylases (deacetylation) and/or ADP-ribosyl transferases (ADP-ribosylation) in the introduction section on line 40. There are other enzyme activities exhibited by certain sirtuins including succinylase/desuccinylase, demalonylase, deglutarylase by SIRT5; demyristoylase by SIRT2, 5, 6; and ADP-ribosylation by SIRT4 and 6. These activities direct different functions of respective sirtuins.

Some of these activities in the context of breast and prostate cancer are abruptly reported in the article by the authors. Succinylase/desuccinylase activities of SIRT5 in breast cancer and prostate cancer cells are reported in lines 535-549 and lines 630-632, respectively. It is appropriate to include these activities exhibited by some members sirtuin family in the introduction section along with deacetylase and ADP-ribosyl transferases activities (second paragraph, line 40). This will allow the readers to acknowledge the role of succinylase/desuccinylase activities of SIRT5 in breast (lines 535-549) and prostate cancer (lines 630-632) cells in the subsequent section of the article.

2. The authors also discussed conflicting roles of some of the small molecule inhibitors of sirtuin activities along with knock-out, knockdown, and overexpression of corresponding sirtuin genes in breast and prostate cancer cells. Additionally, the authors also stated the effect of resveratrol on SIRT1, a natural polyphenol and an activator of sirtuins that inhibits the proliferation of breast cancer cells (lines,157-160).

There is relevant literature related to resveratrol effects on sirtuins in both breast cancer and prostate cancer. Inclusion of some of the relevant articles related to resveratrol is appropriate such as ones with PMID: 27358235; PMID: 26459286; PMID: 23248098

3. There are several long sentences all over the article even though they are grammatically correct.
Shortening the long sentences provides clarity and conciseness particularly when several phrases are added. For example, the sentence that runs from lines 763-767.

4. Regarding the sentence that starts on line 91, delete ‘a’ before role to change to dual role.

Author Response

Dear Reviewer,

Thank you for taking out time to review our article. We appreciate the valuable comments and suggestions, which we believe would further enhance our review article. Kindly find below a point-by-point response to your comments and suggestions (highlighted in red) regarding our article. We hope that the revised version of the manuscript is suitable for publication.

  1. As with other classes of histone deacetylases (class I, II, and IV HDACs), sirtuins (class III HDACs) are involved in several cellular processes affecting different protein targets by deacetylase activity.

However, sirtuins are different from class I, II, and class IV in that some of the sirtuins have additional activities besides deacetylation. The authors have cited sirtuins’ function as NAD-dependent deacetylases (deacetylation) and/or ADP-ribosyl transferases (ADP-ribosylation) in the introduction section on line 40. There are other enzyme activities exhibited by certain sirtuins including succinylase/desuccinylase, demalonylase, deglutarylase by SIRT5; demyristoylase by SIRT2, 5, 6; and ADP-ribosylation by SIRT4 and 6. These activities direct different functions of respective sirtuins.

Some of these activities in the context of breast and prostate cancer are abruptly reported in the article by the authors. Succinylase/desuccinylase activities of SIRT5 in breast cancer and prostate cancer cells are reported in lines 535-549 and lines 630-632, respectively. It is appropriate to include these activities exhibited by some members sirtuin family in the introduction section along with deacetylase and ADP-ribosyl transferases activities (second paragraph, line 40). This will allow the readers to acknowledge the role of succinylase/desuccinylase activities of SIRT5 in breast (lines 535-549) and prostate cancer (lines 630-632) cells in the subsequent section of the article.

Author's response:

Thank you for your very insightful comments and suggestions. The other activities of Sirtuins have been included with appropriate references in the introduction section of the manuscript.

  1. The authors also discussed conflicting roles of some of the small molecule inhibitors of sirtuin activities along with knock-out, knockdown, and overexpression of corresponding sirtuin genes in breast and prostate cancer cells. Additionally, the authors also stated the effect of resveratrol on SIRT1, a natural polyphenol and an activator of sirtuins that inhibits the proliferation of breast cancer cells (lines,157-160).

There is relevant literature related to resveratrol effects on sirtuins in both breast cancer and prostate cancer. Inclusion of some of the relevant articles related to resveratrol is appropriate such as ones with PMID: 27358235; PMID: 26459286; PMID: 23248098

Author's response:

Thank you for your valuable comments and suggestions. We have included and cited evidence of the toxic effect of resveratrol in breast cancer cells to corroborate the study discussed in the section you queried.

  1. There are several long sentences all over the article even though they are grammatically correct. Shortening the long sentences provides clarity and conciseness particularly when several phrases are added. For example, the sentence that runs from lines 763-767.

Author's response:

Thank you for this comment and suggestion. Revisions have been done in this regard.

  1. Regarding the sentence that starts on line 91, delete ‘a’ before role to change to dual role.

Author's response:

We apologize for the typographical error. We have eliminated the “a” and have conducted English language editing and proofreading of the entire manuscript.

Reviewer 2 Report

The manuscript entitled “The Mechanistic Roles of Sirtuins and their Regulation by MicroRNAs in Breast and Prostate Cancer” by Cosmos Ifeanyi Onyiba, Christopher J. Scarlett  and Judith Weidenhofer is a review paper focused on an overview of the rile of mammalian Sirtuins in cancer, specifically in breast and prostate cancers. The authors have done a very good job in collecting the extensive information and the manuscript deserve publication as such a summary of the available data may be useful for the field.

However, the manuscript presents a few aspects that prevent its publication in its present form. These shall be clarified, amended or further improved, for the manuscript to be published.

1) In the Title, the words ‘and their Regulation by MicroRNAs’ should be removed. The role of MicroRNAs for the regulation of Sirtuins is given a too relevant space in the title when, actually, the authors only discuss regulation by miRNA on SIRT1 and SIRT7 in breast cancer and on SIRT1 in prostate cancer.

2) In the Abstract, two lines and a half are dedicated to microRNAs. As mentioned for the title, also in the abstract too much emphasis is given for little data shown. The lines about miRNAs should be replaced by the summary about the localization-mediated control of Sirtuins in the two cancers considered. This is in fact the major focus of the review.

3) In the Introduction section, page 2, line 57, some context about the role of mammalian Sirtuins in diseases shall be provided that introduces the work of the authors on further diseases. This has been recently done. After the references [19-25], please insert the paragraph below:

‘Thus, deregulation of Sirtuins can be involved in perturbations found in metabolic disorders and cancer. Recently, the role of all mammalian Sirtuins in metabolic disorder affecting six human tissues has been reported (Maissan et al., Biology, 2021, 10(3), 194). Furthermore, Sirtuins have been reported to play a dual role …’

4) In the Introduction section, page 2, lines 71 and 72, please provide the percentage corresponding to 18,110 (males) and 19,866 (females). The number do not indicate the incidence in the population.

5) At the end of the Introduction section, page 3, lines 82-83, the authors indicate again ‘… and their regulation by microRNAs (miRNAs) in breast and prostate cancer.’ Before to mention this, the authors shall include in the Introduction – and not later in the text – the rationale that links cancer with miRNA, and the role of the latter in cancer. Because this is the Introduction section, details of the ole of miRNAs in different cancers shall be summarized, before to then indicate why the authors will look at the role of miRNA ‘only’ in breast and prostate cancers – and why the details are just a few for these two cancers. What about the other cancer types: is the data scarce also there?

6) In Figure 2, there is a mistake about SIRT4 localization. The sentence ‘… SIRT4 has also been detected in the cytoplasm.’ Should be corrected with ‘SIRT4 has also been detected in both nucleus and cytoplasm. This is also – correctly – indicated in the text (Introduction, page 2, lines 46-47).

7) Figure 3 should be inserted in the text after mentioning it for the first time, for the reader to visualize an overview of the regulations that will be discussed in detail in the subsequent paragraphs. Currently, Figure 3 is mentioned at page 3, lines 94-95, but it is only shown at page14.

8) ‘LNCap’ appears in the text for the first time at page 7, but it I not defined. The authors should provide the full name here, as it does not appear throughout the text.

9) The information summarized is extensive and of interest. However, a section and a figure are missing where shared mechanisms of Sirtuins on targets and localization of the former in both of breast and prostate cancers are described in detail and shown. This has been indicated by the authors to be a relevant aspect of the review (Introduction section, pages 2-3, lines 78-79 (page 2) and 80-81 (page 3). However, a section where the information about common – in addition to the specific – differences are highlighted is not provided in a comparative, detailed manner. The new Figure that the authors may need to prepare should show both common and different (specific) mechanisms for the two cancer types investigated in the review.

Author Response

Dear Reviewer,

Thank you for taking out time to review our article. We appreciate the valuable comments and suggestions, which we believe would further enhance our review article. Kindly find below a point-by-point response to your comments and suggestions (highlighted in red) regarding our article. We hope that the revised version of the manuscript is suitable for publication.

1) In the Title, the words ‘and their Regulation by MicroRNAs’ should be removed. The role of MicroRNAs for the regulation of Sirtuins is given a too relevant space in the title when, actually, the authors only discuss regulation by miRNA on SIRT1 and SIRT7 in breast cancer and on SIRT1 in prostate cancer.

Author's response:

Thank you for your valuable observation and suggestion regarding the title of our manuscript. We have eliminated the words “and their Regulation by MicroRNAs” to place more emphasis on the mechanistic roles of Sirtuins in breast and prostate cancer.

2) In the Abstract, two lines and a half are dedicated to microRNAs. As mentioned for the title, also in the abstract too much emphasis is given for little data shown. The lines about miRNAs should be replaced by the summary about the localization-mediated control of Sirtuins in the two cancers considered. This is in fact the major focus of the review.

Author's response:

Thank you for your valuable observation and suggestion regarding much emphasis on microRNAs in the abstract. We have eliminated the too much emphasis on microRNAs in the abstract and have added “Specifically, we highlight the involvement of Sirtuins in the regulation of various proteins implicated in proliferation, apoptosis, autophagy, chemoresistance, invasion, migration, and metastasis of breast and prostate cancer” to lay more emphasis on the mechanism of Sirtuins in prostate and breast cancer, which is the major focus of our review.

3) In the Introduction section, page 2, line 57, some context about the role of mammalian Sirtuins in diseases shall be provided that introduces the work of the authors on further diseases. This has been recently done. After the references [19-25], please insert the paragraph below:

‘Thus, deregulation of Sirtuins can be involved in perturbations found in metabolic disorders and cancer. Recently, the role of all mammalian Sirtuins in metabolic disorder affecting six human tissues has been reported (Maissan et al., Biology, 2021, 10(3), 194). Furthermore, Sirtuins have been reported to play a dual role …’

Author's response:

Thank you for this valuable suggestion. We have added this information to the introduction section of the manuscript.

4) In the Introduction section, page 2, lines 71 and 72, please provide the percentage corresponding to 18,110 (males) and 19,866 (females). The number do not indicate the incidence in the population.

Author's response:

Thank you for the valuable suggestion. The incidences (in percentages) have been used to replace the numbers.

5) At the end of the Introduction section, page 3, lines 82-83, the authors indicate again ‘… and their regulation by microRNAs (miRNAs) in breast and prostate cancer.’ Before to mention this, the authors shall include in the Introduction – and not later in the text – the rationale that links cancer with miRNA, and the role of the latter in cancer. Because this is the Introduction section, details of the role of miRNAs in different cancers shall be summarized, before to then indicate why the authors will look at the role of miRNA ‘only’ in breast and prostate cancers – and why the details are just a few for these two cancers. What about the other cancer types: is the data scarce also there?

Authors response:

Thank you for your valuable comments. We have eliminated “….and their regulation by microRNAs (miRNAs)” to ensure that our discussion focuses mainly on the mechanistic roles of Sirtuins in breast and prostate cancer. Moreover, as a promising topic that is yet to be fully explored in cancer, we mechanistically discussed miRNAs as regulators of Sirtuins' activities in breast and prostate cancer, which may be of potential prognostic, diagnostic, and therapeutic relevance. As indicated in Tables 3 and 4, not much has been done in this area, which is among the future research opportunities revealed in this review. Also, we did not summarize the role of miRNAs in different cancers because our focus was breast and prostate cancer.

6) In Figure 2, there is a mistake about SIRT4 localization. The sentence ‘… SIRT4 has also been detected in the cytoplasm.’ Should be corrected with ‘SIRT4 has also been detected in both nucleus and cytoplasm. This is also – correctly – indicated in the text (Introduction, page 2, lines 46-47).

Author's response:

Thank you for this observation. We have made the change in the description for Figure 2.

7) Figure 3 should be inserted in the text after mentioning it for the first time, for the reader to visualize an overview of the regulations that will be discussed in detail in the subsequent paragraphs. Currently, Figure 3 is mentioned at page 3, lines 94-95, but it is only shown at page 14.

Author's response:

Thank you for this observation. Figure 3 has been moved to the appropriate position in the manuscript.

8) ‘LNCap’ appears in the text for the first time at page 7, but it I not defined. The authors should provide the full name here, as it does not appear throughout the text.

Author's response:

Thank this observation. LNCaP has been defined at the first mention as “lymph node carcinoma of the prostate”.

9) The information summarized is extensive and of interest. However, a section and a figure are missing where shared mechanisms of Sirtuins on targets and localization of the former in both of breast and prostate cancers are described in detail and shown. This has been indicated by the authors to be a relevant aspect of the review (Introduction section, pages 2-3, lines 78-79 (page 2) and 80-81 (page 3). However, a section where the information about common – in addition to the specific – differences are highlighted is not provided in a comparative, detailed manner. The new Figure that the authors may need to prepare should show both common and different (specific) mechanisms for the two cancer types investigated in the review.

Author's response:

Thank you for this valuable comment and suggestion. However, we wish to state our inability to include this information based on the following reasons: first, the information regarding the common and different mechanisms between the two cancers reviewed is slightly revealed when you juxtapose figures 3a and 3b but not discussed since it is not our main focus. Second, considering the sparsity of information, the currently available information is insufficient to open discussions regarding this query at this stage of Sirtuins research on prostate and breast cancer. Again, the heterogeneous nature of the limited mechanistic studies on Sirtuins and their target (as shown in figures 3a and 3b) makes it difficult to draw common pathways between both cancers. Third, our main intention is to reveal the currently available information on the topic, with the view of highlighting the grey areas that need to be elaborated, thus instigating the need for future elaborative studies regarding the topic. Notwithstanding, we will keep this suggestion in mind in our future reviews on Sirtuins, depending on how much information is available. Notwithstanding, we have revised the conclusion section of the manuscript by adding more information that generally addresses this point.

Reviewer 3 Report

The review study by Onyiba et al focused on the biological functions of human sirtuins (SIRT1–7) in breast and prostate cancer, which are regulated by microRNAs (miRNAs), and play an important role in diverse biological phenomena such as apoptosis, proliferation, differentiation, epithelial-Mesenchymal transitions...etc.

The article is well written.

I found this review study very interesting and would be very useful for a clinical procedure.

I also found that this study will provide new insights into our understanding of the regulation of the Sirtuins1-7 in breast and prostate cancer.

The methodology is fine and no further control is required.

I found the conclusion to be in line with the evidence and arguments presented.

Tables and Figures are fine.

The references are well updated.

Nice story!!

Author Response

Dear Reviewer,

Thank you for taking out time to review our article. We appreciate the valuable comments and suggestions, which we believe would further enhance our review article. Kindly find below a point-by-point response to your comments and suggestions (highlighted in red) regarding our article.

The review study by Onyiba et al focused on the biological functions of human sirtuins (SIRT1–7) in breast and prostate cancer, which are regulated by microRNAs (miRNAs), and play an important role in diverse biological phenomena such as apoptosis, proliferation, differentiation, epithelial-Mesenchymal transitions...etc.

The article is well written. I found this review study very interesting and would be very useful for a clinical procedure. I also found that this study will provide new insights into our understanding of the regulation of the Sirtuins1-7 in breast and prostate cancer. The methodology is fine and no further control is required. I found the conclusion to be in line with the evidence and arguments presented. Tables and Figures are fine. The references are well updated.

Nice story!!

Author's response:

Thank you for your comments on our article.
